# Understanding Spatiotemporal Variations of Ridership by Multiple Taxi Services

**Wenbo Zhang** [1,*] **, Yinfei Xi** [2] **and Satish V. Ukkusuri** [3]

1    School of Transportation, Southeast University, Nanjing 211189, China
2    Department of Civil Engineering, Monash University, Clayton, VIC 3800, Australia; Yinfei.Xi@monash.edu
3    Lyles School of Civil Engineering, Purdue University, W. Lafayette, IN 47907, USA; sukkusur@purdue.edu
*    Correspondence: wenbozhang@seu.edu.cn; Tel.: +86-025-8379-2500

**Abstract:** Recent years have seen the big growth of app-based taxi services by not only competing for rides with street-hailing taxi services but also generating new taxi rides. Moreover, the innovation in dynamic pricing also makes it competitive in both passenger and driver sides. However, current literature still lacks better understandings of induced changes in spatiotemporal variations in multiple taxi ridership after app-based taxi service launch. This study develops two study cases in New York City to explore impacts of presence of app-based taxi services on daily total and street-hailing taxi rides and impacts of dynamic pricing on hourly app-based taxi rides. Considering the panel data and treatment effect measurement in this problem, we introduce a mixed modeling structure with both geographically weighted panel regression and difference-in-difference estimator. This mixed modeling structure outperforms traditional fixed effects model in our study cases. Empirical analyses identified the significant spatiotemporal variations in impacts of presence of app-based taxi services; for instance, impacts daily total taxi rides in 2014 and 2016 and impacts on street-hailing taxi rides from 2012 to 2016. Moreover, we capture the spatial variations in impacts of dynamic pricing on hourly app-based taxi rides, as well as significant impacts of time of day, day of week, and vehicle supply.

**Keywords:** app-based taxi services; treatment effects; geographically weighted panel regression; taxi ridership

## 1. Introduction

Recent years have seen big increases in app-based taxi services (or ride-hailing services such as Uber and Lyft) across the world. For example, the app-based taxi ridership doubled annually over the last few years to 133 million passengers in 2016 and is approaching street-hailing (or the traditional yellow taxicabs) ridership level in New York City (NYC) [1]. Note that, the travelers in NYC cannot book traditional street-hailing cabs by smartphone-based applications. This emerging mobility provides similar door-to-door services as traditional street hailing taxicabs, yet is with higher quality of service. First, the smartphone-based application provides various products that can meet needs of different groups and, more importantly, can show you basic market information around request locations, such as nearby vehicles and estimated waiting time. Second, the dynamic pricing, integrating with free entry market model, can attract as many driver partners as possible during rush hours to serve and shift portion of demand to other transportation modes. Both innovations in operations contribute to the big increases, however, also challenge the dominance of street-hailing taxicabs in cities. The declines of daily street-hailing taxicab ridership in NYC have increased from 5% in 2014 to 50% in 2016, compared with daily ridership before Uber launch in 2012 [1,2]. Similar declines and even bankruptcy of street-hailing taxicab companies are observed in US major cities, for instance,

30% in Austin [3], bankruptcy in Chicago [3], 22% decline and bankruptcy in Boston [4], and even 65% and bankruptcy in San Francisco [5].

In addition to the statistics of demand shift or net demand increase at city or system level, there still lacks empirical studies on the process at a smaller scale. Since taxi rides are highly influenced by urban social and economic activities and its distribution is spatially unbalanced, as well as spatially correlated. Simple analyses at city or system level will lead to misspecifications in differences across regions (also called spatial heterogeneity). Moreover, we should have a better understanding of app-based taxi interventions, including how many net ride increases are induced by app-based taxi services and where are those net rides from, how many ride decreases are in street-hailing taxicabs across years and regions, are surge pricing (one pricing strategy used by Uber that we may be charged at a higher fare rate) effective in reducing demand during rush hours and balancing demand and supply, and how surge pricing settings influence app-based taxi rides. All these practical problems are challenging while urban transportation policymakers develop appropriate regulation policies for taxi industry.

In addition, two main difficulties in methodologies make this much more challengeable, other than obtaining high-resolution ride datasets of both street-hailing and app-based taxis. The first difficulty is the treatment effect measurement. The object of interest is a comparison of all taxi and street-hailing taxicab ridership (or app-based taxi ridership) for the same geographical unit when presence, and when no presence, of app-based taxi services (or surge pricing). The problem is that we can at most observe ridership under only one level of treatment. Within the field of treatment effect measurement, difference-in-difference (DID) approach is one typical solution, which relies on the presence of additional data in the form of samples of treated and control units before and after treatment [6,7]. Combing with time effects or panel data, DID estimator is an efficient spatiotemporal framework within which to evaluate the impact of changing amenities over time [8]. More importantly, the DID estimator can be fully integrated intro econometric regressions, which makes it more popular in related analyses.

The second difficulty is the spatial autocorrelations among taxi rides. In general, the taxi rides in one unit are related to characteristics of not only the unit but also the immediate neighbors of the units, for instance the phenomena of spillover. Traditional global regression and spatial autoregressive model cannot specify these interactions. Although spatial autoregressive model takes neighboring outcomes into consideration, the basic assumption underlying this approach is that the relationships of interest are stationary or homogeneous spatially. Recent advances in local regression modeling, such as geographically weighted regression (GWR), provides a new perspective through a repeated estimation of a local regression at each point in space with a subsample of cross sectional data properly weighted according to their proximity to each regression point. GWR addresses the non-stationary distribution across units.

This study is developed to investigate the app-based taxi service interventions on multiple ridership. To track the treatments over time and address spatial heterogeneity and autocorrelations among taxi rides, we utilize the extensions of GWR to panel data, then integrate with DID estimators to measure treatment effects. Two real cases in NYC are applied to the modeling structure, including the long-term effect of presence of app-based taxi on daily total taxi (i.e., both app-based and street hailing) and street-hailing taxicab ridership and the short-term effect of surge pricing on hourly app-based taxi ridership. The main contributions are two-fold: First, we integrate DID estimator with GWR for panel data, which allows us to measure treatment effects, as well as address non-stationary spatial distribution; and Second, this study is one of the first few studies on taxi ridership at different scales and its spatiotemporal variations. The following sections are organized as follows: Section 2 reviews current literature on taxi ridership specification and advances in treatment effects, as well as spatial heterogeneity; Section 3 presents the mixed modeling structure and estimation methods; Section 4 introduces two case studies; and Section 5 explores model performance and empirical findings; and Section 6 summarizes the study and indicates future study.

## 2. Literature Review

### 2.1. Taxi Ridership Estimation

Ubiquitous mobile computing and massive data it generates present new opportunities to advance our understandings of taxi rides at both spatial and temporal scales. Zhang et al. [9] introduced zero-inflated negative binomial model for hourly street hailing taxicab ridership, considering the nature of count data and excessive of zero observations, and explored the impacts of time of day, day of week, socioeconomic, land use, and built environment. Kamga et al. summarized impacts of time of day, day of week, and weather based on statistics of daily taxi ridership [10]. Yang and Gonzales introduced an ordinary least square model to specify taxi ridership and estimated the impacts of population, age income, education, transit access time, and employment [11]. Qian and Ukkusuri quantified the spatial correlations in taxi ridership using a geographically weighted regression model and revealed the impacts of demographic, land use, transit accessibility, and commuting time [12]. With real dataset from Uber (i.e., one of the largest app-based taxi service provider), Cohen et al. figured out price elasticities in app-based taxi services by relating request ratio and dynamic pricing and identified the discontinuity (sharp decrease) in app-based taxi service purchase at 1.25 [13]. Overall, most existing studies focused on street-hailing taxicab system in which the availability of huge ride record has a better spatial and temporal coverage. However, the widely adopted global econometric models failed to specify spatial heterogeneity. Although there are very few discussions on impacts of surge pricing on app-based taxi service purchase, we have not seen empirical discussions on combination of both street-hailing and app-based taxi ridership, as well as inclusion of spatial heterogeneity in impacts of surge pricing.

### 2.2. Treatment Effects and Spatial Heterogeneity

In general, modeling spatial heterogeneity are in two manners: discrete and continuous. The discrete manner is based on predefined spatial units that are usually considered within the regression as fixed, random, or nested effects. Fixed effects are presented as spatial indicators for regions and can vary over space. However, introducing a large number of fixed effects will result in insufficient observations within each spatial unit for parameter estimations. Random effects can be approximated as a weighted average of the mean of observations in the spatial units. However, the estimates tend to be same as global mean for small sub sample sizes and tend towards the unpooled dummy estimations for large sub sample sizes. Nested effects first group spatial units and generalize random effects model to more than one hierarchical level in which parameters are estimated by a probability model. The continuous manner, including polynomial regression and locally weighted regression, is directly driven by dataset while modeling of parameter instability, instead of exogenous assumptions concerning spatial units. In polynomial regression, parameters can vary as a function of the coordinates but estimates tend to get distorted at edges of study area. Moreover, the modeling structure is too smooth for modeling local variations. The locally weighted regression performs a series of weighted least squares regressions on subsamples, where the influence of an observation decreases with the proximity to a regression point.

GWR is one most presentative form of locally weighted regression and has been approved to be efficient in modeling spatial heterogeneity. In different problem settings, GWR has many variations, including adaptive bandwidth for weighting, mixed GWR, geographically weighted panel regression (GWPR), and geographically and temporally weighted regression (GTWR). Helbich et al. developed mixed geographically weighted regression with both global variables and local variables to specify hedonic house price [14]. Liu et al. [15], as well as Ge et al. [16] and Zhang et al. [17], proposed the modeling structure of GTWR that introduces a weighting function with both spatial and temporal proximities and allows to specify spatiotemporal non-stationarity. However, it is challengeable to obtain better spatial and temporal proximities and merge both proximities into one reliable weight matrix. Bruna and Yu also verified that GTWR are not comparable with the estimates that can be obtained

under a local approach to panel data estimation [18]. For panel data structure, Yu [19] and Cai et al. [20] developed a GWPR modeling structure that performs panel data models on weighted variables at each location. Other than model variations, Leong and Yue proposed conditional geographically weighted regression to modify traditional geographically weighted regression with an optimal bandwidth, instead of iterative approaches of checking Akaike's information criterion [21].

As one popular treatment effect measurement method, DID estimator is usually included into econometric regression structures. Kondo et al. included DID estimator into a Poisson random effects model to explore the effects of abandoned building remediation strategy on crimes [22]. Dube et al. developed spatial difference-in-difference approach for impacts of public mass transit on house prices, by integrating spatial autoregressive and did estimator based on differences in dependent and independent variables. Impacts of public mass transit on house prices [23]. Chagas et al. introduced a similar spatial DID structure for impacts of sugarcane production on respiratory diseases [24]. Delgado and Florax demonstrated the spatial DID model performance with Monte Carlo simulation [25]. Instead of integrations with spatial autoregressive model, GWR is an alternative for treatment effect estimation across spatial units, which has been applied in the effects of greening vacant land on property values [26].

## 3. Methodology

### 3.1. Causal Inference

Measuring the app-based taxi intervention on ridership is a comparison of all taxi and street-hailing taxicab ridership (or app-based taxi ridership) for the same geographical unit when presence, and when no presence, of app-based taxi services (or surge pricing), as shown in equation1. However, we can at most observe ridership under only one level of treatment in real world. Thus, Equation (1) is not estimable. Equations (2) to (4) shows the alternative estimations. Equation (2) is designed based on cross-sectional dataset and assumes spatial unit homogeneity. Equation (3) is the standard mathematical form of DID with both treatment and control groups. Equation (4) is simplified form of equation 3 without control group and assumes temporal homogeneity. In our study, app-based taxi services launched in the study area at almost same time. Thus, we do not have control group in this area. Moreover, the spatial autocorrelations and heterogeneity in taxi rides are significant, which may violate the assumption of Equation (2). In final, we choose Equation (4) as an estimator for treatment effect. The assumption of temporal homogeneity in Equation (4) likely hold in a short time series, e.g., few years.

$$\Delta_i = Y^T_{i,t_0} - Y^T_{i,t_0} \tag{1}$$

$$\hat{\Delta}_i = Y^T_{i,t_0} - Y^C_{j,t_0} \tag{2}$$

$$\hat{\Delta}_i = \left(Y^C_{i,t_1} - Y^T_{i,t_0}\right) - \left(Y^C_{j,t_1} - Y^C_{j,t_0}\right) \tag{3}$$

$$\hat{\Delta}_i = Y^T_{i,t_1} - Y^T_{i,t_0} \tag{4}$$

where $\Delta_i$ is the real treatment effect at spatial unit $i$; $\hat{\Delta}_i$ is the estimator for treatment effect at spatial unit $i$; $T$ is the treatment group; $C$ is the control group; $i$ and $j$ are spatial units; $t_1$ is the periods after treatment; $t_0$ is the periods before treatment; $Y$ is the observed or estimated ridership.

### 3.2. Geographically Weighted Panel Regression

The DID estimator can be further integrated with econometric regressions. Here, we choose GWPR as Equation (5) and estimate treatment effects as Equation (6). The GWPR is the GWR for panel data, which can address spatial non-stationary, estimate local coefficients, and track treatment effects over time. The model estimation will perform at each spatial unit through introducing a weighting function that can include influencing factors from neighboring spatial units, as shown in

Equations (7) and (8). The common weighting function is the Gaussian kernel (shown in Equation (9)) that can account for distance decay effects and is utilized in different fields including taxi ridership estimation [12]. In Gaussian kernel, the key step is to propose an optimal bandwidth that is the threshold of weighting. Here, we will choose an adaptive bandwidth other than fixed bandwidth for all spatial units, which can account for local variations and improve model performance.

$$Y_{it} = \alpha_i + \sum_{k=1}^{K} \beta_{ik} x_{itk} + \gamma_i t + \delta_{it}(p_{it} \times D_i) + \varepsilon_{it} \tag{5}$$

$$\hat{\Delta}_{it} = E\left(Y_{it}^T | D_i = 1, X_{it} = X_{it}\right) - E\left(Y_{it}^T | D_i = 0, X_{it} = X_{it}\right) = \delta_{it} \tag{6}$$

$$\hat{\beta}_i = (X W_i X)^{-1} X^T W_i Y \tag{7}$$

$$W_i = diag\left(w_{i1}, w_{i2}, \cdots, w_{ij}, \cdots, w_{in}\right) \tag{8}$$

$$w_{ij} = \begin{cases} exp\left[-\frac{1}{2}\left(\frac{d_{ij}}{b_i}\right)^2\right] & , d_{ij} < b_i \\ 0 & , \text{otherwise} \end{cases} \tag{9}$$

where $Y_{it}$ is the ridership at spatial unit $i$ in time of $t$; $p_{it} \in \{0,1\}$ is the post treatment period dummy, $D_i \in \{0,1\}$ is the treatment dummy, $\delta_{it}$ is the treatment effect at spatial unit $i$ in time of $t$; $\beta_{ik}$ is the local coefficient for variable $k$ at location $i$; $K$ is the number of explanatory variables; $w_{ij}$ is the weigh for spatial unit pair of $i$ and $j$; $d_{ij}$ is the spatial distance between two spatial units $i$ and $j$; and $b_i$ is the bandwidth for spatial unit $i$.

### 3.3. Estimation

We refer to an iterative approach for GWPR estimation that can yield time-invariant local coefficients from panel data and generate adaptive bandwidth. The main idea behind the iterative approach is that subsample data at each location, weight explanatory variables, apply panel data estimation, and iterate over above steps.

1) Select a local bandwidth $b_i$ and generate weighting matrix $W_i$ at spatial unit $i$;
2) Subsample observed data for local estimation at spatial unit $i$;
3) Weight all observations of $j$'s variables with weighting matrix $W_i$;
4) Apply fixed effect model to the weighted subsample data;
5) Iterate over step 1 to 4 for an optimal local bandwidth through checking Akaike Information Criterion (AIC);
6) Iterate over step 1 to 5 for all spatial units.

## 4. Data and Case Design

The study area is the New York City, shown in Figure 1, which is the largest taxi market in the North America and is operating the largest street-hailing taxicabs fleet over 13,000. The daily street-hailing taxicab ridership is about 330,000 in 2017; although, it is reduced by almost half compared with its peak records. On the other hand, the app-based taxi services are popular in NYC with more than 5 platforms. The fleet of the largest app-based taxi service provider (i.e., Uber) has outnumbered the street-hailing taxicabs in 2015. The Uber ridership is approaching the daily street-hailing taxicab ridership level in 2017. Moreover, NYC has perfect open datasets including street-hailing taxicab ride records from 2009 to 2016, Uber ride records from April to September, 2014 (ref: https://www1.nyc.gov/site/tlc/about/tlc-trip-record-data.page) and socioeconomic dataset from American Community Survey (ACS), which can support our empirical analyses on treatment effects.

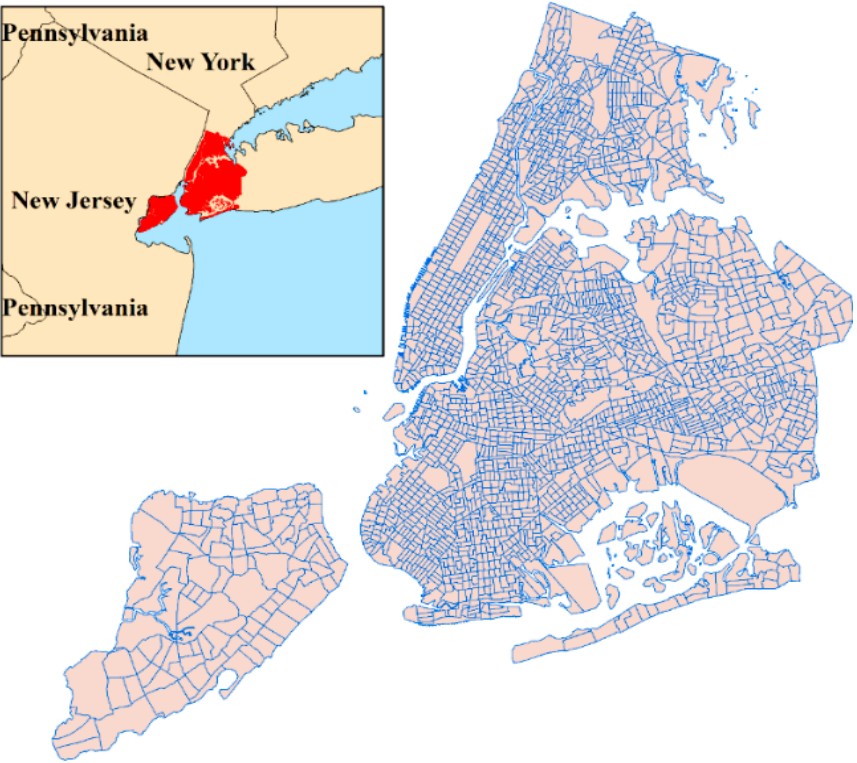

**Figure 1.** Study area and census tracts.

Except for the current available dataset, we also develop a tool for data extraction from Uber Mobile platform (a web version of Uber client app, ref: https://m.uber.com). This tool is similar to Chen et al. [27]. However, we extend collection to the whole NYC and collect data from 8 April to 1 May 2017 (Note that observations on April 12 and 14 are removed due to incomplete data collection). The raw Uber dataset in 2017 is the empty vehicle trajectory points every 2 to 3 s and the surge pricing multiplier of 470 predefined locations across NYC. Note that the straight distance between two predefined locations varies from 600 m to 1500 m, depending on their spatial locations. During the empty vehicle trajectory dataset, we only focus on the drivers of interest who operated in at least 10 days during observation and filter out 43,000 active drivers that is similar as Uber fleet in NYC. From the trajectory of these targeted active drivers, we analyze the trajectory chains and discern Uber rides through identifying trajectory intervals greater than 60 s but less than 2 h, as well as greater than 400 m displacement, which are measured from statistics of historical taxi trip records.

Instead of working directly with massive ride start points in terms of longitude and latitude, they are aggregated at census tracts that are small subdivisions of a city and provide a stable set of geographic units for presentation of statistical data. This process can enable us to explore spatial variations of treatment effects and other influential factors. In NYC, there are about 2164 census tracts, more than 70% of which have the area less than 0.1 square miles (similar as a 500 m by 500 m grid), as shown in Figure 1. Furthermore, the 470 predefined locations are matched with census tracts if it locates within one census tract. Thus, we can have the surge pricing multipliers at level of census tracts. In final, we develop two study cases to explore impacts of app-based taxi services on multiple ridership, including case 1: long-term effects of presence of app-based taxi services on street-hailing and total (i.e., both street-hailing and app-based) taxi daily ridership, and case 2: short-term effects of dynamic pricing on hourly app-based taxi ridership.

### 4.1. Case 1: Long-Term Effects of Presence of App-Based Taxi Services on Daily Ridership

This case is designed to explore the ride increases or decreases induced by app-based taxi service launch and their spatiotemporal variations. From available datasets, we can develop two subcases, case 1-a for impacts on total rides of both street-hailing and app-based taxis and case 1-b for impacts on only street-hailing rides. Limited by data availability, we implement few preprocesses: (1) subsample 22 days street-hailing daily ridership that can match Uber data collection periods by day of week, and similar preprocess is applied to 2014 Uber ride records; (2) assume almost no app-based rides between 2012 (Uber launch year) and 2013, considering the growth stage and data availability; (3) assume temporal homogeneity of app-based rides between 2016 and 2017; thus, total ridership in 2016 is the summation of street-hailing ridership in 2016 and app-based ridership in 2017; and (4) remove year 2015 in case 1-a, since that year has noneligible app-based rides yet we do not have app-based ride records. In final, we have 22 daily ridership for all 2164 census tracts from 2009 to 2016 for case 1-b and same temporal scale but excluding 2015 for case 1-a.

In case 1-a, we deploy two treatments, including presence of app-based taxi services in 2014 and in 2016. In case 1-b, we deploy five treatments, including presence of app-based taxi services from 2012 (launch year) to 2016. In addition to treatment variables, both the cases also share few explanatory variables, such as indicator variable of weekend, density of population aged between 22 and 35, density of individual workers with commuting time over 45 min, density of individual workers with commuting time less than 15 min, density of occupied household units, density of household without vehicles, and density of population under poverty line. These explanatory variables indicate the levels of resident wealth, demographics, vehicle ownership, and commuting characteristics at each census tract. Table 1 summarizes the statistics of these variables and presents pairs of explanatory variables with the Pearson correlations over 0.5, as well as VIF value for multicollinearity test. Although there are few Pearson correlations up to 0.7, we still keep all explanatory variables considering very low VIF values.

### 4.2. Case 2: Short-Term Effects of Dynamic Pricing on Hourly Ridership

This case is designed to understand how dynamic pricing affect passenger side. Due to lack of dynamic pricing information in 2014, we only focus on impacts in 2017. We have 19 observation hours for all 2164 census tracts and 22 days from 8 April to 1 May 2017. In real, the dynamic pricing is mainly achieved through multiplying base fare rate by a surge multiplier varying from 1.0 to around 3.0 with an increment of 0.1. However, we cannot introduce each of such many levels of surge multiplier as one treatment variable. Moreover, the duration of each level of surge multiplier is different in each hour, which increases difficulties while deploying treatment variables for dynamic pricing. To overcome, we refer to a surge multiplier threshold of 1.25 where is the discontinuity (sharp decrease) of app-based taxi demand curve, approved theoretically [28] and empirically [13]. Moreover, we collect duration of surge multiplier greater than 1.2 in each hour and deploy two treatment variables based on quantiles of duration distribution. Other than, we also introduce few explanatory variables, for instance, day of week, time of day, street-hailing taxicab supply, and app-based taxi supply. Our high-resolution Uber trajectory dataset can approximate levels of app-based taxi supply in an accurate manner. However, street-hailing taxicab supply is estimated through counting ride drop-offs for one hour in each census tract, since street-hailing ride records only provide origin and destination information yet no any trajectory points. The summary statistics, as well as Pearson correlations and Variance Inflation Factor (VIF) values, are presented in Table 1(c). All Pearson correlations are less than 0.5 and VIF values are very small, around 1.

**Table 1.** Summary statistics of variables for all three cases.

| Variable | Description | Min | Max | Mean (Standard Deviation) or Percentage | Correlations (Absolute Value Greater than 0.5) | VIF |
|---|---|---|---|---|---|---|
| **(a) Case 1-a** | | | | | | |
| DTR | Daily Total Ridership of both app-based and street-hailing | 0 | 14,213 | 217.61 (830.21) | - | - |
| WK | Indicator variable of weekends, 1-if weekends, 0-if weekdays | 0 | 1 | 63.64%/36.36% | - | 1.00 |
| TR14 | Treatment indicator of Uber in 2014 | 0 | 1 | 85.71%/14.29% | - | 1.07 |
| TR16 | Treatment indicator of Uber in 2016 | 0 | 1 | 85.71%/14.29% | - | 1.07 |
| AGE | Density of population aged between 22 and 35 (100,000 per square mile) | 0 | 1032.28 | 110.94 (103.38) | NV (0.6944) *; TT45 | 2.57 |
| TT45 | Density of individual workers with commuting time over 45 min (10,000 per square mile) | 0 | 5325.23 | 830.49 (641.37) | AGE; NV | 1.91 |
| TT15 | Density of individual workers with commuting time less than 15 min (10,000 per square mile) | 0 | 4404.97 | 223.26 (278.69) | - | 1.18 |
| OU | Density of occupied household units (10,000 per square mile) | 0 | 11,964.11 | 633.35 (897.86) | - | 1.07 |
| NV | Density of household without vehicles (10,000 per square mile) | 0 | 9551.84 | 1125.53 (1282.60) | AGE; TT45; UP | 2.68 |
| UP | Density of population who is under poverty line (10,000 per square mile) | 0 | 1119.12 | 104.34 (122.21) | NV | 1.73 |
| **(b) Case 1-b** | | | | | | |
| DYR | Daily Street-hailing Taxicab Ridership | 0 | 13743 | 207.92 (811.29) | - | - |
| WK | Indicator variable of weekends, 1-if weekends, 0-if weekdays | 0 | 1 | 63.64%/36.36% | - | 1.00 |
| TR12 | Treatment indicator of Uber in 2012 | 0 | 1 | 87.5%/12.5% | | 1.23 |
| TR13 | Treatment indicator of Uber in 2013 | 0 | 1 | 87.5%/12.5% | | 1.24 |
| TR14 | Treatment indicator of Uber in 2014 | 0 | 1 | 87.5%/12.5% | - | 1.25 |
| TR15 | Treatment indicator of Uber in 2015 | 0 | 1 | 87.5%/12.5% | | 1.27 |
| TR16 | Treatment indicator of Uber in 2016 | 0 | 1 | 87.5%/12.5% | - | 1.26 |
| AGE | Density of population aged between 22 and 35 (100,000 per square mile) | 0 | 1032.28 | 111.76 (103.76) | NV (0.6713) *; TT45 | 2.18 |
| TT45 | Density of individual workers with commuting time over 45 min (10,000 per square mile) | 0 | 5325.23 | 837.51 (643.78) | AGE; NV | 1.12 |
| TT15 | Density of individual workers with commuting time less than 15 min (10,000 per square mile) | 0 | 4404.97 | 222.58 (278.17) | - | 1.64 |
| OU | Density of occupied household units (10,000 per square mile) | 0 | 11,964.11 | 614.10 (865.60) | - | 1.17 |
| NV | Density of household without vehicles (10,000 per square mile) | 0 | 9551.84 | 1130.10 (1284.04) | AGE; TT45; UP | 2.12 |
| UP | Density of population who is under poverty line (100,000 per square mile) | 0 | 1119.12 | 105.32 (123.02) | NV | 1.36 |

**Table 1.** *Cont.*

| Variable | Description | Min | Max | Mean (Standard Deviation) or Percentage | Correlations (Absolute Value Greater than 0.5) | VIF |
|---|---|---|---|---|---|---|
| **(c) Case 2** | | | | | | |
| HUR | Hourly App-based Ridership | 0 | 399 | 2.17 (5.01) | - | - |
| WK | Indicator variable of weekends, 1-if weekends, 0-if weekdays | 0 | 1 | 63.64%/36.36% | - | 1.05 |
| T6-T23$ | Hour indicator of 6am to midnight | 0 | 1 | 94.74%/5.26% | - | <1.50 |
| TR24 | Treatment indicator of surge multiplier greater than 1.2 for more than 24 min in one hour | 0 | 1 | 99.15%/0.85% | - | 1.01 |
| TR624 | Treatment indicator of surge multiplier greater than 1.2 for more than 6 min but less than 24 min in one hour | 0 | 1 | 98.42%/1.58% | - | 1.03 |
| TS | Hourly available yellow taxicabs | 0 | 973 | 8.24 (32.58) | - | 1.19 |
| US | Hourly available Uber vehicles | 0 | 598 | 11.93 (15.66) | - | 1.09 |

'-': not available; '*' indicate the maximum Pearson correlation value; '$' includes a set of 18 h indicator variables from 6 am to midnight, we present in one row and common statistics shared by all hour indicator variables.

## 5. Empirical Findings

### 5.1. Model Performance

All cases are with typical panel data structures. We also compare our proposed model estimation with traditional fixed effects model for panel data. First, we extract residuals from two fixed effects model including fixed time effects and fixed time and location effects. The local spatial autocorrelation tests by Moran's I are applied to the average residual in each census tract. Figure 2a presents the census tracts with significant spatial autocorrelations among average residuals. For both fixed effects model and all cases, there are about 400 to 500 census tracts with significant spatial autocorrelations. Especially in fixed time effects model for case 2, it shows much more census tracts with significant spatial autocorrelations. Comparing two fixed effects model, it seems introducing fixed location effects cannot address spatial autocorrelations in case 1, as opposed to conditions in case 2. Figure 2b shows the spatial distribution of average residuals by GPWR estimation. In all three cases, there are almost no significant clusters of similar residuals. The spatial autocorrelations can be well addressed by GWPR. Table 2 summarizes the performance of all estimated models, including log-likelihood, R squared value, AIC, and AICc (i.e., AIC with a correction for finite sample sizes). The GWPR model outperforms traditional fixed effects model, which is with much higher log-likelihood, higher R squared value, and much lower AIC and AICc values. In addition, the introduction of fixed location effects cannot improve the specification by fixed time effects model.

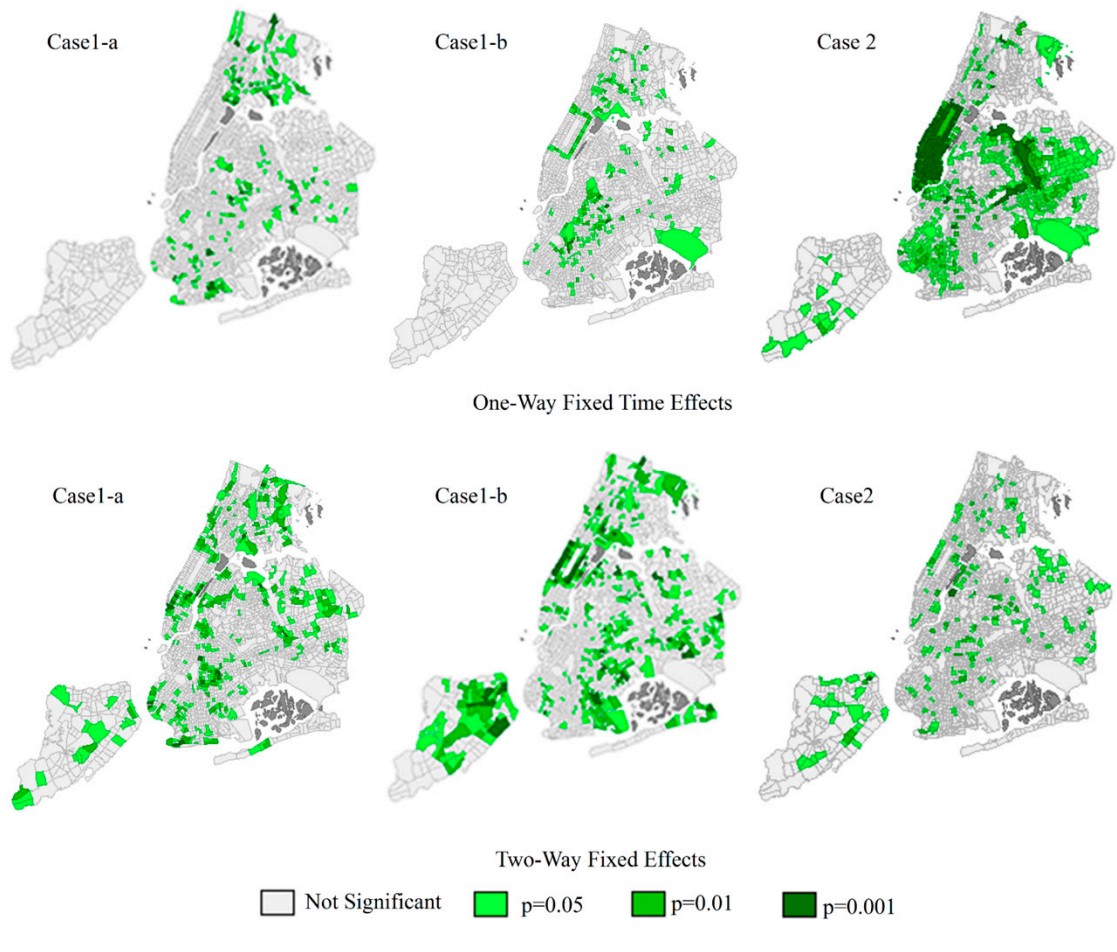

(**a**) Spatial autocorrelations in residuals

**Figure 2.** *Cont.*

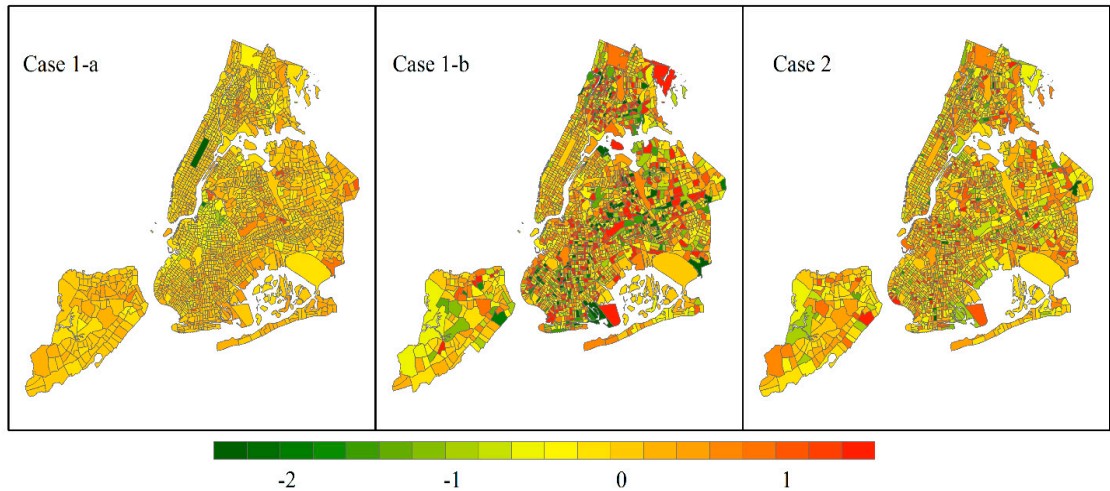

(**b**) Residual distribution by geographically weighted panel regression (GWPR).

**Figure 2.** Spatial autocorrelations among residuals.

**Table 2.** Model performance of panel models and GWPR.

| Model Performance | Case 1-a | Case 1-b | Case 2 |
|---|---|---|---|
| No. of Observations | 333,256 | 380,864 | 904,552 |
| Log-likelihood only with intercept | −521,120 | −376,563 | −1,053,514 |
| *One-way fixed time effect model* | | | |
| Log-likelihood | −386,112 | −369,626 | −972,428 |
| Degree of freedom | 10 | 13 | 25 |
| $R^2$ | 0.555 | 0.034 | 0.164 |
| AIC | 772,244 | 739,278 | 1,944,907 |
| AICc | 772,244 | 739,278 | 1,944,907 |
| *Two-way fixed effect model* | | | |
| Log-likelihood | −386,112 | −399,292 | −972,363 |
| Degree of freedom | 2173 | 2176 | 2188 |
| $R^2$ | 0.555 | 0.036 | 0.164 |
| AIC | 776,570 | 743,604 | 1,949,112 |
| AICc | 776,598 | 743,629 | 1,949,122 |
| *Geographically weighted panel regression* | | | |
| Log-likelihood | −237,284 | −231,354 | −864,634 |
| Degree of freedom | 10,820 | 21620 | 50,472 |
| $R^2$ | 0.864 | 0.279 | 0.294 |
| AIC | 496,209 | 505,949 | 1,830,212 |
| AICc | 496,935 | 508,551 | 1,836,184 |

*5.2. Case 1 Impacts of Presence of App-Based Taxi Services*

Figure 3a presents the local treatment effects of presence of app-based taxi services on total taxi ridership. In year of 2014, the presence of app-based taxi service induced few new demands by a lower level of 1 to 2% mainly in an outer Borough (i.e., Brooklyn) downtown and the remote airport (i.e., JFK). There are almost no induced demands in city core areas (i.e., Manhattan). Although we can observe declines of millions of street-hailing rides in 2014, the demand just shifted from street-hailing to app-based services. In contrast, the condition of net increase in total taxi ridership has extended to almost all boroughs other than city core areas in 2016. More importantly, the increase rates are around 3% to 4%, higher than before. The decreases in total taxi ridership are significant in city core areas other than one point of interest (i.e., Central Park). Two main reasons may contribute to these declines: First, the city core areas are developing better shared bike system that is an alternative transportation

mode for short trips; and second, street-hailing taxicabs may serve more rides outside of city core areas confronting with high competitions with app-based taxi services in city core areas.

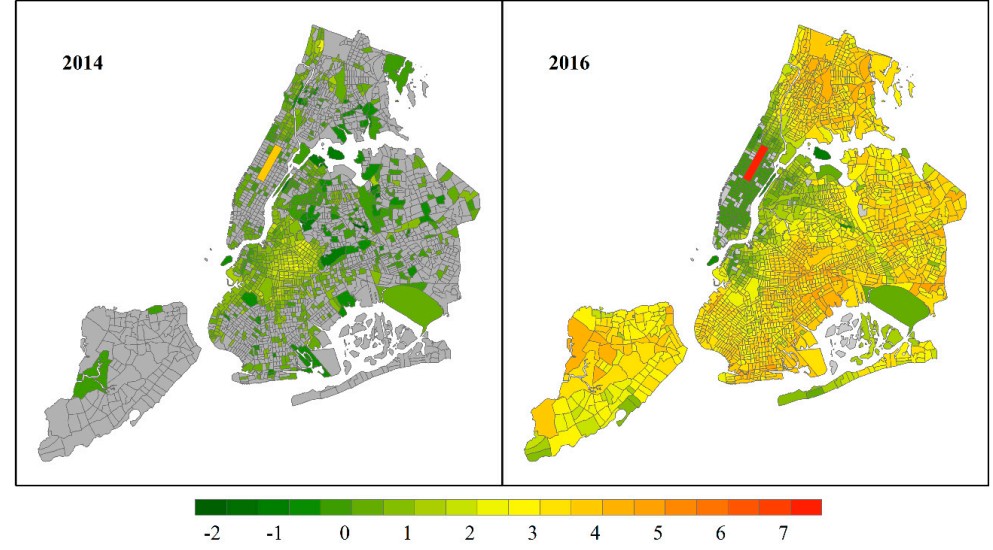

(**a**) Impacts on total taxi ridership in years of 2014 and 2016

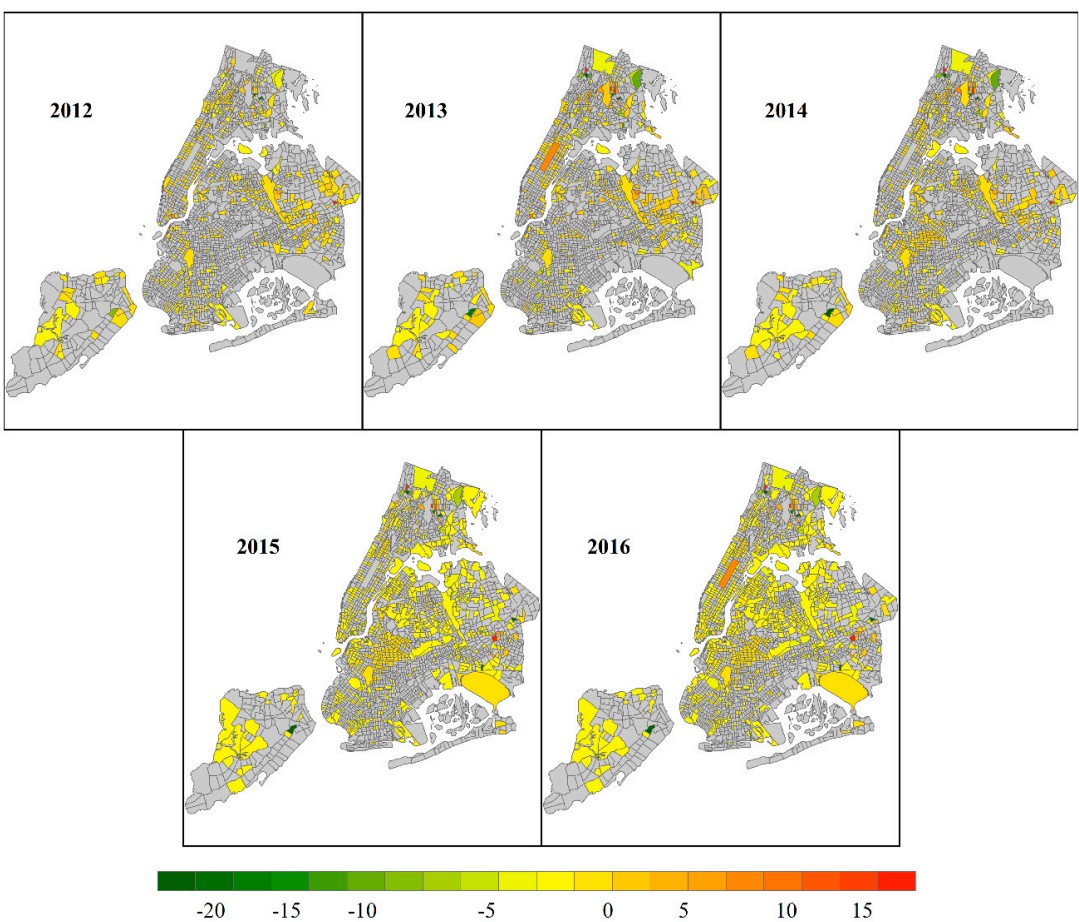

(**b**) Impacts on street-hailing taxicab ridership in years of 2012 to 2016

**Figure 3.** Impacts of presence of app-based taxi services. (Note grey regions mean the corresponding impacts are statistically insignificant).

Figure 3b shows the local treatment effects of presence of app-based taxi services on street-hailing rides from launch year of app-based taxi services. Although app-based taxi services launched in 2012, it did not affect street-hailing rides until 2014, excluding few remote areas. From 2014, we can see significant declines of street-hailing rides in remote areas and city core areas. In addition, one interesting point is the significant decreases in city core areas but increases in the outer Brough downtown identified in case 1-a, which indicates the street-hailing taxicabs are moving out city core areas for services and supports our explanations on declines of total rides. From 2015, street-hailing taxicab rides at the remote airport are also increased by 2 to 3%. However, presence of app-based taxis did not influence street-hailing taxicab rides at another airport close to city core areas.

## 5.3. Case 2 Impacts of Dynamic Pricing

Figure 4a presents the impacts of two treatment effects on hourly app-based ridership. Most areas cannot observe significant impacts, likely due to the frequency of surge pricing. Figure 4b shows how many minutes one census tract is with higher surge multiplier (>1.2). App-based taxicab platforms rarely issue higher surge multiplier and more than half of census tracts are almost without or very low surge multiplier. Even in city core areas always with high travel demand, more than 80% of observation minutes are without higher surge multiplier. Although there are few census tracts with significant treatment effects, we can still explore few interesting findings. First, the high surge multiplier for more than 24 min in one hour will decrease app-based taxi rides by almost 1% at the remote airport. This is likely since rides from the remote airport are always distant trips with a higher base fare and higher multiplier will make trip costs unaffordable. In contrast, a shorter duration of higher surge multiplier will not affect hourly ridership at same location. Another interesting point is that even with longer duration of higher surge multiplier, hourly ridership surprisingly increases by a small rate in city core areas and few remote areas. This is likely because the higher surge multiplier may attract more driver partners and improve request success ratio. Moreover, most short trips with smaller base fares happen in city core areas. Higher surge multiplier will not increase trip cost a lot, and app-based taxi services may be the only option for travel in remote regions thus higher multiplier will not influence hourly rides.

Figure 5 presents the impacts of both street-hailing and app-based taxi supply on hourly app-based rides. Unsurprisingly, more available app-based vehicles can increase hourly app-based rides by around 0.1%. Moreover, the impacts are spatially homogeneous. Regarding the impacts of street-hailing taxi supply, they are significant mostly in the city core areas. The finding reveals the potential high competitions in the city core areas where high hourly app-based taxi rides are along with higher available street-hailing taxicabs.

Figure 6a shows the impacts of day of week on hourly app-based rides. In the city core areas full of workplaces, hourly app-based rides are expected to decline by at least 0.2% on weekends. However, potential more travels by flights on weekends contribute to at least 0.2% more app-based taxi rides. Another significant increase in hourly app-based taxi rides on weekends locate outside of city core areas where may be not covered well by other transportation modes. Figure 6b demonstrates the impacts of time of day on hourly app-based rides. First, compared with daytime, hourly app-based taxi rides may decrease by more than 2% during late night in areas generally with high travel demand. Considering airport operation schedules, it does make sense that airports have higher hourly app-based taxi rides during daytime. The city core areas are expected to have 1% more hourly app-based taxi rides between 4 and 5 p.m., as well as evening peak, other than morning peak hours and off-peak hours. The increased app-based taxi rides from 4 p.m. are likely due to street-hailing taxicab shift and temporary shortage of available street-hailing taxicabs. Another interesting point is that more census tracts with decreased app-based rides after early morning and morning peak. Combining with hourly app-based rides in city core areas by time of day, we may confirm the existence of considerable part-time driver partners outside of city core areas. Each day, they share commuting trips with others from houses to core city areas and share back trips to houses after work in core city areas.

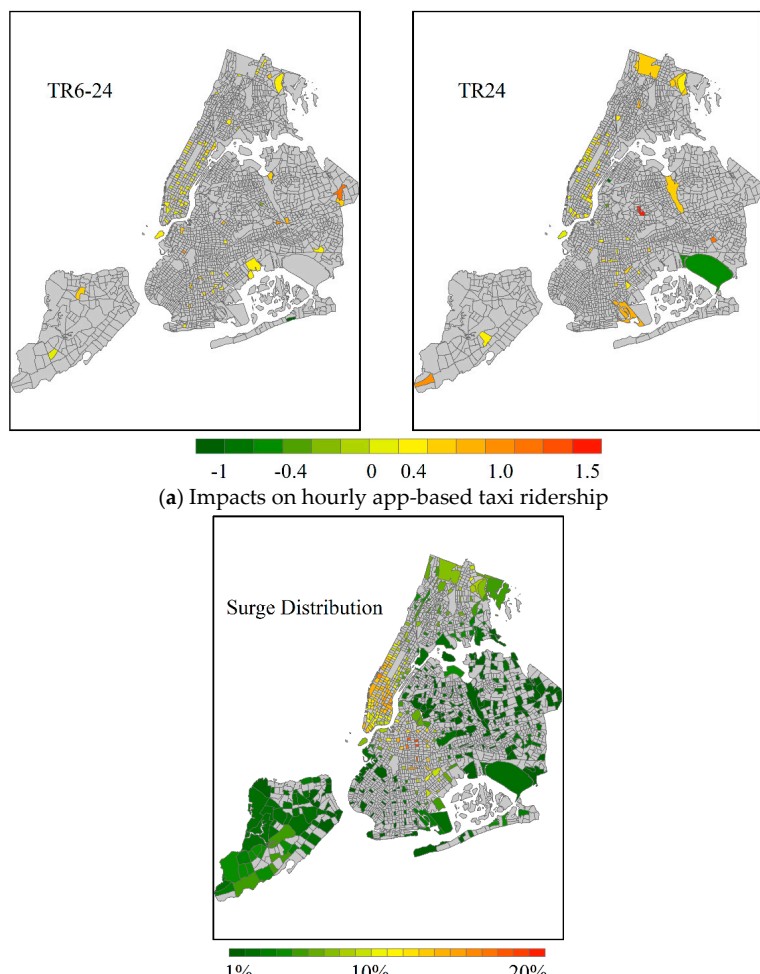

(**a**) Impacts on hourly app-based taxi ridership

(**b**) Durations in terms of percentage of all 25,080 min

**Figure 4.** Impacts of durations of higher surge multiplier (>1.2). (Note grey regions mean the corresponding impacts are statistically insignificant).

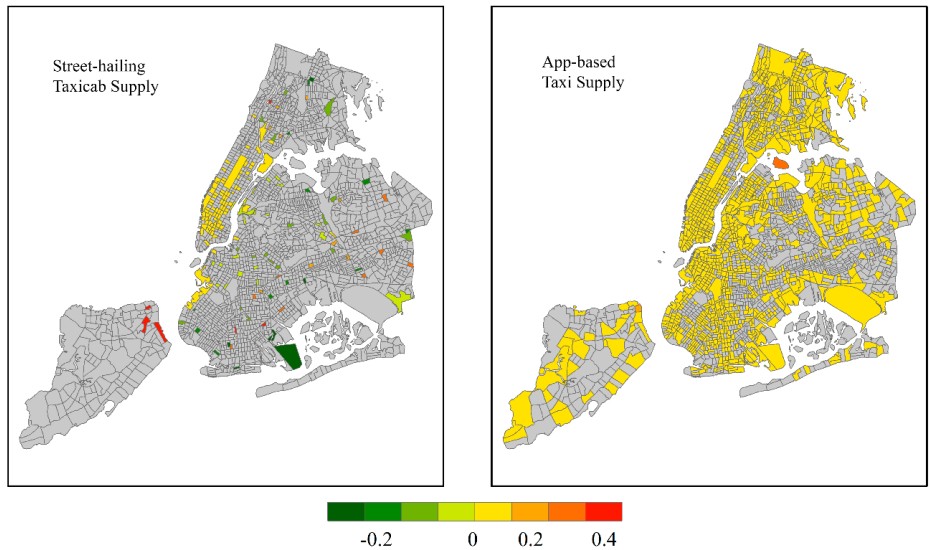

**Figure 5.** Impacts of vehicle supply on hourly app-based taxi ridership. (Note grey regions mean the corresponding impacts are statistically insignificant).

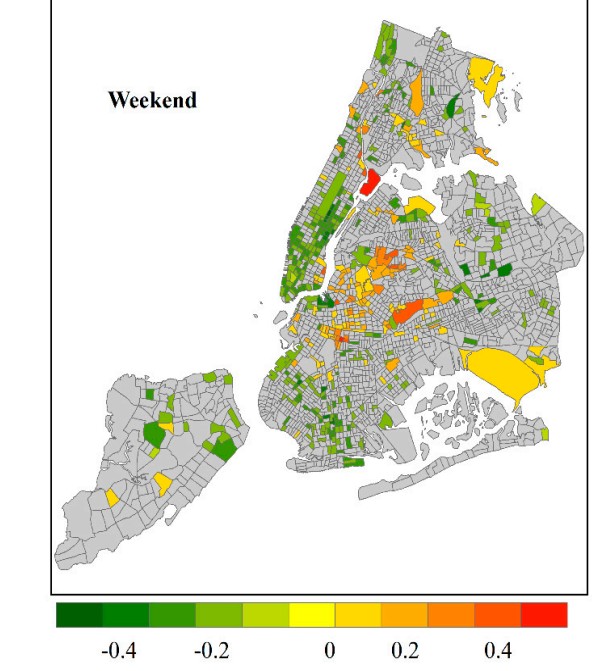

(**a**) Impacts of day of week on hourly app-based rides

(**b**) Impacts of time of day on hourly app-based rides

**Figure 6.** Impacts of temporal characteristics on hourly app-based taxi ridership. (Note grey regions mean the corresponding impacts are statistically insignificant).

## 6. Conclusions

In this study, we leverage the advances in both treatment effect measurements and geographically weighted regression models to measure the impacts of presence of app-based taxi services on total and street-hailing taxi ridership, as well as impacts of surge pricing on app-based taxi ridership. The adaption of geographically weighted regression models for panel data not only address spatial non-stationary but also capture treatment effects over times. We develop two real cases to explore aforementioned impacts respectively. Both cases verify geographically weighted panel regression outperform traditional fixed effects model. The modeling structure addresses two difficulties in exploring such complex spatiotemporal datasets, which are the quantifications of both treatment effects and spatial heterogeneity.

The empirical results identify the spatiotemporal variations in impacts of presence of app-based taxi services on multiple ridership. Presence of app-based taxi services generates very few new taxi rides in few regions outside city cores in 2014 but leads to much more new rides in almost all regions outside city cores in 2016. Moreover, its impacts on street-hailing taxi rides are not significant until 2015, about three years after app-based taxi service launch. To avoid high competitions with app-based taxi services, street-hailing taxicabs are gradually moving outside city cores for rides. We also identify the negative impacts of long duration of higher surge multiplier on hourly app-based rides at airports but interestingly positive impacts at city cores and few remote regions. In addition, vehicle supply, as well as time of day and day of week, contribute to hourly app-based ride changes. All these empirical analyses are valuable for urban transportation policymakers while developing taxi regulation policies.

As one of the first few empirical studies on spatiotemporal variations in multiple taxi ridership, the geographically weighted panel regression with DID estimator can capture treatment effects of app-based taxi services over times. However, this study is limited by data availability. We still lack of real app-based taxi ride records, which is solved by approximations from trajectory dataset in this study. This may underestimate the number of app-based taxi rides considering we cannot distinguish few rides from empty vehicle trajectory points. Furthermore, lack of latest street-hailing taxicab ride records in 2017, when the Uber data is collected, leads to assumptions of temporal homogeneity between 2016 and 2017. However, this can be easily addressed once new dataset releases. Last, the impacts of surge pricing, as well as public transit, on hourly app-based taxi rides should have in-depth discussion once multi-source transportation big data is available. Such exploration will lead to better understanding of urban transportation mode shift.

**Author Contributions:** Conceptualization, W.Z.; methodology, W.Z. and S.V.U.; software, W.Z. and Y.X.; validation, W.Z.; formal analysis, W.Z.; writing—original draft preparation, W.Z. and S.V.U.; writing—review and editing, W.Z.; visualization, W.Z. and Y.X. All authors have read and agreed to the published version of the manuscript.

**Funding:** This research was funded by National Natural Science Foundation of China—Youth Program, grant number 52002064, and Humanities and Social Science Youth Program, Ministry of Education, China, grant number 20YJC630216. And the APC was funded by Southeast Unviersity.

**Conflicts of Interest:** The authors declare no conflict of interest.

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
