# Peer review of "Understanding Spatiotemporal Variations of Ridership by Multiple Taxi Services"

_ijgi, doi:10.3390/ijgi9120757_

Round 1

Reviewer 1 Report

The increasing number of app-based taxi services such as Uber has led to increased competition in the taxi market, especially in cities. To investigate the impact of app-based taxi services and their dynamic pricing models on taxi rides, the authors conduct a data-driven approach using location-based data and empirically evaluate their impact. As a sample, they use both open data and data from the app Uber in New York City. For this purpose, the authors use a combination of geographically weighted panel regression and a difference-in-difference estimator. First, they conclude that the modeling structure used exceeds the traditional model with fixed effects.

Moreover, they discover that app-based taxi services lead to very few new rides in a few regions outside the city centers in 2014, but many more new rides in almost all regions outside the city center in 2016. They also notice that the impact on traditional taxi rides is not significant until 2015, three years after introducing Uber. Furthermore, they identify negative effects of a long-term increased surge multiplier on hourly app-based rides at airports and positive effects on city centers and few remote regions.

The authors conduct an interesting study, which has not yet been conducted before in such a configuration and research design. The research subject is interesting and relevant, and the methodology is well implemented and described.

There are, however, some crucial aspects that need to be critically reviewed and should be revised:

The introduction of the paper motivates the research approach mainly based on a research gap. However, the motivation should also be driven by practical and theoretical considerations since the reader does not realize to whom and, more importantly, how the study offers added value to practice and research. Besides, more literature references should anchor the theoretical concepts and the study in general within the existing literature.

The description of the theoretical background (chapter 2) is quite limited and should be improved. The topics app-based taxi services with corresponding examples of platforms and dynamic pricing should be introduced in the introduction. Adding definitions and explanations would help the reader access the topic clearly and pick up nonspecialist readers. Moreover, important underlying concepts, such as spatial heterogeneity, are not sufficiently defined or explained. The authors enter at a very high level of knowledge and hardly pick up the reader.

The description of the methodology is very detailed, but, like the concepts mentioned above, it enters at a very high subject-specific level. The geographically weighted panel regression and difference-in-difference methods should be defined and explained before describing this study's approach. The data collection procedure is clearly stated, but the authors should mention the data sources of the open data. The authors explain their procedure in detail, but occasionally they lack justifications, for instance, the definition of the intervals (p. 6, lines 4-5), the derivation of the variables (p. 7, lines 5-9), or why the days April 12 and 14 are excluded from the analysis (p. 7, line 19). Also, the argument 'due to [...] space limits' should not be used in a journal article. The abbreviation VIF is not explained throughout the text, which makes it challenging to understand the tables. The tables do not reveal the meaning of the numbers in the brackets, as there is no legend or hint in the caption. Extending the annotations below the tables would make interpretation easier for the reader.

There are considerable deficiencies in the map presentations. Each of them contains a scale with different values. These values, however, are not labeled, which makes an interpretation very difficult. The authors should introduce a scale bar as a cartographic element and labels for the legends/scales. Figure 2a contains a green legend, but the green colors are challenging to distinguish. It would be helpful for the reader if these colors could be distinguished more clearly.  Besides, some maps are blurred. The other illustrations' scales are colored from green to red and thus not barrier-free what makes it impossible for readers with visual disorders to interpret them. It would be appropriate to choose a different barrier-free color scheme.

Additionally, it is not clear what the grey parts of the maps represent, and for this reason, it should be listed in the legend. The image captions are also confusing, as the reader does not know what they refer to. An example is Figure 4a, which is entitled 'impacts of hourly app-based taxi ridership.' Here, the caption should indicate what exactly influences taxi rides. Since the authors often refer to the airport, since significant spatial results can be derived here, it would be useful to mark it on the maps.

The conclusion is relatively short compared to the other parts of the paper and should be considered more. The evaluation of the chosen analytical methods is clearly explained and evaluated. Unfortunately, as in chapter 1, the research design's practical and scientific relevance is not clear. The authors should abstract their model and illustrate the relevance for practice and theory more clearly.

Reviewer 2 Report

For future research, include more recent data and consider other aspects that lead to a multi-criteria analysis.

Author Response

Great thanks for your comments and recommendations. We have included that into the future direction at the end of section 'conclusion'. In addition, we also carefully check the manuscript and correct some minor spelling/grammar mistakes. 

Reviewer 3 Report

This paper presents an analysis of spatio-temporal changes of taxi use in NYC after the introduction of app-based ridehailing services through a GTWR model. The study is interesting and well written. I have some minor suggestion to provide to the authors in order to improve it:

  • I think that since the very beginning definitions of appbased services and street-hailing services should be provided. The paper is currently a bit confusing, but if I got it clear, street-hailing services are traditional taxi services, while appbased ones are inew ridehailing services such as Uber, Lyft, Via... I think this clarification must be stated in the introduction or Methods section, since NYC is a unique case and the fact that traditional taxis cannot be booked by app should be highlighted.
  • The authors could add to their references a recent study published in IJGI: Zhang, X., Huang, B., & Zhu, S. (2020). Spatiotemporal Varying Effects of Built Environment on Taxi and Ride-Hailing Ridership in New York City. ISPRS International Journal of Geo-Information9(8), 475.
  • The authors should revise their use of language in the paper, since some sentences might sound awkward
  • In future researchers it would be interesting to evaluate the same impacts of appbased ridehailing services on Public Transport/Subway use. This could be added in conclusion section.

Reviewer 4 Report

Work is well written with no edits needed.

Author Response

The authors really appreciate your efforts while reviewing. During the round of minor revision, we also carefully check our manuscript and correct some minor spelling and grammar mistakes. 

This manuscript is a resubmission of an earlier submission. The following is a list of the peer review reports and author responses from that submission.